# Encapsulation of Photothermal Nanoparticles in Stealth and pH-Responsive Micelles for Eradication of Infectious Biofilms In Vitro and In Vivo

**DOI:** 10.3390/nano11123180

**Published:** 2021-11-24

**Authors:** Ruifang Gao, Linzhu Su, Tianrong Yu, Jian Liu, Henny C. van der Mei, Yijin Ren, Gaojian Chen, Linqi Shi, Brandon W. Peterson, Henk J. Busscher

**Affiliations:** 1College of Chemistry, Chemical Engineering and Materials Science, Soochow University, Suzhou 215006, China; r.gao@umcg.nl; 2University of Groningen, University Medical Center Groningen, Department of Biomedical Engineering, 9713 AV Groningen, The Netherlands; sulinzhu@mail.nankai.edu.cn (L.S.); Tianrongyu@yeah.net (T.Y.); h.c.van.der.mei@umcg.nl (H.C.v.d.M.); b.w.peterson@umcg.nl (B.W.P.); 3Key Laboratory of Functional Polymer Materials of Ministry of Education State, Key Laboratory of Medicinal Chemical Biology, Institute of Polymer Chemistry College of Chemistry, Nankai University, Tianjin 300071, China; 4Jiangsu Key Laboratory for Carbon-Based Functional Materials & Devices, Institute of Functional Nano & Soft Materials (FUNSOM), Soochow University, Suzhou 215006, China; jliu@suda.edu.cn; 5University of Groningen, University Medical Center Groningen, Department of Orthodontics, 9713 AV Groningen, The Netherlands; y.ren@umcg.nl

**Keywords:** biofilm, infection, polydopamine, photothermal nanoparticles, micelles, self-targeting

## Abstract

Photothermal nanoparticles can be used for non-antibiotic-based eradication of infectious biofilms, but this may cause collateral damage to tissue surrounding an infection site. In order to prevent collateral tissue damage, we encapsulated photothermal polydopamine-nanoparticles (PDA-NPs) in mixed shell polymeric micelles, composed of stealth polyethylene glycol (PEG) and pH-sensitive poly(β-amino ester) (PAE). To achieve encapsulation, PDA-NPs were made hydrophobic by electrostatic binding of indocyanine green (ICG). Coupling of ICG enhanced the photothermal conversion efficacy of PDA-NPs from 33% to 47%. Photothermal conversion was not affected by micellar encapsulation. No cytotoxicity or hemolytic effects of PEG-PAE encapsulated PDA-ICG-NPs were observed. PEG-PAE encapsulated PDA-ICG-NPs showed good penetration and accumulation in a *Staphylococcus aureus* biofilm. Penetration and accumulation were absent when nanoparticles were encapsulated in PEG-micelles without a pH-responsive moiety. PDA-ICG-NPs encapsulated in PEG-PAE-micelles found their way through the blood circulation to a sub-cutaneous infection site after tail-vein injection in mice, yielding faster eradication of infections upon near-infrared (NIR) irradiation than could be achieved after encapsulation in PEG-micelles. Moreover, staphylococcal counts in surrounding tissue were reduced facilitating faster wound healing. Thus, the combined effect of targeting and localized NIR irradiation prevented collateral tissue damage while eradicating an infectious biofilm.

## 1. Introduction

Photothermal nanoparticles convert light to heat [1,2] and are considered for the control of tumors and infectious biofilms, which are becoming more and more difficult through the increasing occurrence of antibiotic-resistant bacterial strains [3]. The use of photothermal nanoparticles for infection-control has its own advantages and disadvantages. One of the major advantages is the broad-spectrum bacterial killing by photothermal nanoparticles on a non-antibiotic basis. However, compared to photothermal treatment of centimeter-sized tumors, irradiation of a micrometer-sized infection site will usually cause collateral tissue damage due to the heat generated by photothermal nanoparticles.

Near-infrared (NIR) irradiation is preferred for photothermal treatment due to its “deep” penetration in tissue that is clinically still confined to around 5 to 10 mm maximally [4]. Different materials such as carbon quantum dots [5], graphene [6], noble metals [7], copper chalcogenide nanomaterials [8], polyaniline [9] and polydopamine [10] nanoparticles (PDA-NPs) have shown the ability to generate heat upon NIR irradiation, among which PDA-NPs are easy to prepare and possesses excellent biocompatibility and biodegradability [11]. Growth of infectious biofilms in vitro in the presence of photothermal PDA-NPs yielded incorporation of the nanoparticles in a biofilm and the heat generated by subsequent NIR irradiation killed bacterial inhabitants of the biofilm [12]. This suggests potential of PDA-NPs for prophylactic, clinical use to prevent post-surgical infections. PDA-NPs, however, were unable to penetrate and accumulate in existing, infectious biofilms and kill its bacterial inhabitants without causing collateral tissue damage, which excludes their therapeutic use [13].

Penetration and accumulation of photothermal NPs in an infectious biofilm after blood-injection can be enhanced by suitable surface modification, but this requires tedious chemistry, often at the expense of biocompatibility [12,14]. Encapsulation of hydrophobic antimicrobials in pH-responsive mixed shell polymeric micelles, composed of a combination of poly(ethylene glycol) (PEG) and poly(β-amino ester) (PAE) [15,16,17], has shown to be effective in stimulating penetration and accumulation of antimicrobials in an infectious biofilm. PEG provides the micelles with stealth properties enabling them to be transported in the blood circulation, while the response of PAE in the acidic environment of a biofilm creates a more positive charge of PEG-PAE micelles than at a physiological pH of 7.4. These features facilitate self-targeting, penetration, and accumulation through electrostatic double-layer attraction with negatively charged bacteria in a biofilm [15]. Unfortunately, hydrophilic PDA-NPs [18,19] cannot be encapsulated in PEG-PAE micelles unless the nanoparticles are made hydrophobic.

This paper describes the coating of PDA-NPs with indocyanine green (ICG) to make hydrophobic PDA-ICG-NPs that can be encapsulated in stealth, pH-responsive PEG-PAE micelles. ICG was employed as it also possesses photothermal properties [20,21] and is an FDA approved NIR dye [22]. Merits of PDA-ICG/PEG-PAE micelles with respect to the photothermal eradication of an existing infectious biofilm (therapeutic-mode) will be compared in vitro and in vivo with those of PDA-NPs encapsulated in single shell, PEG micelles, lacking the pH-adaptive feature.

## 2. Materials and Methods

### 2.1. Materials

3-Hydroxytyramine hydrochloride (98%), NH_4_OH (28–30%), indocyanine green (ICG), potassium hydrogen phosphate, potassium dihydrogen phosphate, sodium hydroxide, and agar were purchased from Sigma-Aldrich (Shanghai, China). Tryptone Soya Broth (TSB; OXOID, Basingstoke, UK), RPMI-1640 medium, Fetal Bovine Serum, penicillin, streptomycin, trypsin-EDTA, SYTO^®^9 and Cell Counting Kit-8 (CCK-8) were obtained from Thermo Fisher Scientific, Inc. (Beijing, China). Ethanol, hydrochloric acid, dimethyl sulfoxide and tetrahydrofuran were obtained from Sinopharm Chemical Reagent Co., Ltd. (Shanghai, China). All aqueous solutions were prepared in 18.2 MΩ cm purified Milli-Q water (Millipore, Bedford, MA, USA). All chemicals were of analytical grade and used without further purification.

### 2.2. Synthesis of PDA-NPs and ICG Coating

The synthesis of PDA-NPs has been previously reported [12,23], as briefly described in the Appendix A. PDA-NPs were modified with ICG by mixing 2 mL of a PDA-NP suspension in water (1 mg/mL) with 1 mL of ICG in water (1 mg/mL). The pH of the mixture was adjusted to 1.85 with hydrochloric acid, after which the mixture was sonicated for 2 min and stirred for 3 h at room temperature, which resulted in electrostatic binding between ICG and PDA-NPs. After 3 h, PDA-ICG-NPs were washed three times in deionized water by centrifugation (11,000 g, 5 min) to remove the free ICG and finally washed once in tetrahydrofuran. All synthesis steps involving ICG were conducted in the dark.

### 2.3. Encapsulation of PDA-ICG-NPs in Micelles

For micellar encapsulation, 2 mg of PDA-ICG-NPs were suspended in 1400 μL of dimethyl sulfoxide, while equal amounts (5 mg) of PCL-*b*-PAE and PEG-*b*-PCL [24] were dissolved in 600 μL of tetrahydrofuran and added to the nanoparticle suspension. Formation of PDA-ICG/PEG-PAE micelles with encapsulated PDA-ICG-NPs was done by dripping 2 mL of the mixed nanoparticle suspension/surfactant solution into 6 mL of acetate buffer (pH 3.5) at a rate of 60 μL/min using an injection pump under sonication (Transonic TP 690, ELMA, Germany, 160 W, 35 kHz). Sonication was continued for 20 min. The resulting micelle suspension was dialyzed for 24 h to remove tetrahydrofuran and dimethyl sulfoxide (Mw = 18,000 Da; Suke Jingyi Instrument Co. Ltd., Huai’an, China) and diluted to 10 mL (concentration 0.2 mg/mL) and stored in the dark at 4 °C. PDA-ICG/PEG micelles were prepared using a solution of only PEG-*b*-PCL. 

Nile red loaded micelles were prepared by forming micelles with encapsulated nanoparticles after addition of 100 μL Nile red in dimethylformamide (1 mg/mL) into the nanoparticle suspension/surfactant solution, as described above.

### 2.4. Characterization and Photothermal Efficiency of Nanoparticles before and after Micellar Encapsulation 

For imaging, PDA-NPs and PDA-ICG-NPs were suspended in water (0.5 mg/mL) and 10 μL of each suspension was dropped on a carbon-coated copper grid. After air-drying at room temperature, samples were imaged using a transmission electron microscope (TEM; G-200, Hitachi, Tokyo, Japan) operated at 200 kV.

UV−vis absorption spectra of PDA-NPs, PDA-ICG-NPs and encapsulated nanoparticles were obtained over the wavelength range from 400 nm to 900 nm using a microplate reader (Thermo, Varioskan Flash, Vantaa, Finland). To this end, nanoparticles were suspended in water (0.2 mg/mL).

Diameters of the nanoparticles prior to and after micellar encapsulation were measured using Dynamic Light Scattering (DLS, Malvern ZetaSizer ZS2000, Worcestershire, UK) in phosphate buffered saline (PBS; 10 mM potassium phosphate and 150 mM NaCl) at pH 7.4. Zeta potentials were measured using the same instrument in PBS with pH adjusted to 7.4 and 5.0 by addition of sodium hydroxide and/or hydrochloric acid, to mimic physiological pH conditions and pH conditions in a biofilm, respectively.

Photothermal conversion efficiency of PDA-NPs, PDA-ICP-NPs and encapsulated nanoparticles were determined using nanoparticle suspensions in PBS (0.2 mg/mL). Suspensions (250 μL) were added to 96-wells plates, irradiated with a NIR laser (FCLSR808-FC, Ningbo FINGCO optoelectronic CO. LTD, Ningbo, China) at 808 nm and a power density of 1300 mW/cm^2^ for different periods of time up to 600 s, previously found optimal for bacterial killing and to prevent excessive heat generation that might cause collateral tissue damage [12]. Suspension temperatures were measured using an IR imaging sensor (225S-L24, FOTRIC, Shanghai, China).

Photothermal conversion efficiency was subsequently calculated by setting up an energy balance [12,25], according to
(1)$$\left(m_{H_{2}O}C_{p,H_{2}O}\right)\frac{dT}{dt}={\dot{Q}}_{s}+{\dot{Q}}_{NPs}-{\dot{Q}}_{loss}$$
in which $m_{H_{2}O}$ and $C_{P,H_{2}O}$ are the mass and specific heat of water, respectively, *t* is time and *T* is the suspension temperature. ${\dot{Q}}_{s}$ is the heat uptake per unit time associated with the light absorbed by the suspension fluid. ${\dot{Q}}_{loss}$ is heat loss of the system per unit time. ${\dot{Q}}_{NPs}$ is the photothermal heat generated by nanoparticles per unit and derived from the NIR absorption spectrum according to
(2)$${\dot{Q}}_{NPs}=I\left(1-10^{-A_{808\;nm}}\right)\eta$$
in which I is the laser power and $A_{808\;nm}$ is the absorbance of materials at the wavelength of 808 nm. At equilibrium,
(3)$${\dot{Q}}_{s}={\dot{Q}}_{loss}=hA\Delta T_{max,H_{2}O}$$
in which *h* is the heat transfer coefficient, *A* is the surface area of the system exposed to its surrounding and $\Delta T_{max,H_{2}O}$ is the maximal temperature change of water at equilibrium. Accordingly, the photothermal conversion efficiency η can be derived as
(4)$$\eta =\frac{hA\left(\Delta T_{max,suspension}-\Delta T_{max,H_{2}O}\right)}{I\left(1-10^{-A_{808\;nm}}\right)}$$

In addition, the photothermal stability of PDA-ICG/PEG-PAE micelles was evaluated by subjecting a suspension (0.2 mg/mL, 250 μL) to five cycles of heating (5 min irradiation at 1300 mW/cm^2^) and cooling (5 min at room temperature).

### 2.5. Bacterial Culturing and Harvesting

Bioluminescent *Staphylococcus aureus* Xen36 (PerkinElmer Inc., Waltham, MA, USA) was used from a frozen stock (−80 °C) in 7% (*v*/*v*) DMSO. For experiments, *S. aureus* Xen36 was grown aerobically overnight on blood agar plates supplemented with 200 µg/mL kanamycin at 37 °C. Next, a single colony from a blood agar plate was inoculated in 10 mL TSB with 200 µg/mL kanamycin and grown for 24 h. The entire sub-culture was subsequently transferred into 200 mL TSB with kanamycin and after 17 h incubation bacteria were harvested by centrifugation for 5 min at 5000 g, washed twice in PBS, resuspended in PBS and sonicated three times for 10 s in an ice/water bath to obtain single bacteria (Transonic TP 690, ELMA, Singen, Germany, 160 W, 35 kHz). Suspensions were made in PBS at a bacterial concentration of 3 × 10^8^ bacteria/mL, as determined in a Bürker-Türk counting chamber.

### 2.6. Killing of Planktonic S. aureus by Photothermal Nanoparticles before and after Micellar Encapsulation

To determine the targeting and bacterial killing effects of nanoparticles after micellar encapsulation, 2.5 μL of a *S. aureus* Xen36 suspension in PBS (3 × 10^8^ bacteria/mL) was mixed with 250 μL of a suspension of PDA-ICG/PEG-PAE or PDA-ICG/PEG micelles in PBS (pH 5.0 or 7.4) in 96-wells plates. After 2 h at 37 °C, the mixed suspension was irradiated with a NIR laser (808 nm) at a power density of 1300 mW/cm^2^ for 10 min. Subsequently, suspensions were serially diluted and plated on TSB agar plates. After overnight incubation at 37 °C, the number of colony-forming units (CFU) was enumerated. All experiments were conducted in triplicate with bacteria cultured from three different bacterial cultures.

### 2.7. Biofilm Penetration and Accumulation of Photothermal Nanoparticles after Micellar Encapsulation

First, a *S. aureus* Xen36 biofilm was grown by adding 200 μL *S. aureus* Xen36 suspension in PBS (1 × 10^9^ bacteria/mL) on a 0.7 cm × 0.7 cm glass slide in a 48-wells plate for 1 h at 37 °C to allow bacterial adhesion. Next, the bacterial suspension was removed and glass slides were washed once with PBS to remove the non-adhering bacteria. Subsequently, 500 μL of fresh TSB was added to allow the adhering staphylococci to grow into a biofilm (37 °C at 48 h), while refreshing the medium after 24 h. After 48 h of growth, biofilms were washed once with PBS (pH 5.0 or 7.4) and exposed to 200 μL of a suspension with Nile red loaded PDA-ICG/PEG-PAE or PDA-ICG/PEG micelles (0.5 mg/mL). After 60 min at 37 °C, samples were removed and biofilms washed once with PBS. Next, biofilms were stained with SYTO^®^9 (3 μL SYTO^®^9 in 1 mL water) for 20 min in the dark and examined using a Zeiss CLSM 800 (ZEISS, Jena, Germany). Fluorescence images were collected at 450–570 nm (SYTO^®^9) using a 488 nm laser for excitation and 450−700 nm (Nile red) using a 561 nm laser. Images were taken at 3 μm distances across the depth of a biofilm and analyzed using built-in CLSM software (Zen imaging software 2011, Jena, Germany). Penetration and accumulation of Nile red loaded micelles across the depth of a biofilm was expressed as a red fluorescence intensity relative to the total red and green fluorescence intensity, calculated with the aid of Fiji Image J software (NIH Research Services Branch, Bethesda, MD, USA). Each experiment was carried out in triplicate with separately grown biofilms.

### 2.8. Hemolysis and Cytotoxicity of Photothermal Nanoparticles after Micellar Encapsulation

For the evaluation of hemolytic effects of photothermal nanoparticles before and after micellar encapsulation, 1.2 mL of blood was collected through the eye of three healthy female BALB/c mice and blood of each mouse was transferred into separate vacuum blood collection tubes containing ethylenediamine tetraacetic acid tripotassium salt. After mixing, red blood cells were harvested by centrifugation at 2000 rpm for 10 min, followed by washing with PBS for three times, after which erythrocytes were resuspended in PBS to a volume fraction of 5%. Next, 200 μL of a red blood cell suspension was combined with 1 mL of a photothermal nanoparticle suspension in PBS at different concentrations up to 0.5 mg/mL and left to stand for 3 h at room temperature. Water was used as a positive control (100% hemolysis) and PBS without nanoparticles was used as a negative control (0% hemolysis). Next, suspensions were centrifuged at 2000 rpm for 5 min and the absorbance of the supernatant was measured at 540 nm using a microplate reader. The percentage of hemolysis was expressed as
(5)$$\mathrm{Hemolysis}\;\%=\left[A_{suspension}-A_{PBS}/A_{H_{2}O}-A_{PBS}\right]\times 100\%$$
in which *A* refers to the absorbance of the different suspensions, PBS and water at 540 nm.

For evaluation of cytotoxicity, L929 fibroblasts from the American Type Culture Collection (ATCC-CRL-2014, Manassas, VA, USA) were taken from frozen stock and grown in 75 cm^2^ tissue-culture polystyrene flasks in RPMI-1640 medium supplemented with 10% Fetal Bovine Serum (FBS, Gibco, Shanghai, China), 100 U/mL penicillin (Genview) and 100 μg/mL streptomycin (Solarbio) at 37 °C in 5% CO_2_. Culture media were changed every two days. After grown to 70–80% confluence, fibroblasts were detached from the cell-culture flasks by trypsinization and collected by centrifugation at 1200 rpm for 5 min. Cells were re-suspended in fresh medium at a concentration of 4 × 10^4^ cells/mL, as determined using a Bürker-Türk counting chamber. Next, 200 μL of the cell suspension, i.e., 8000 cells, was put in 96-well plates and incubated at 37 °C in 5% CO_2_. After 12 h, growth medium was removed and 100 μL of a nanoparticle suspension in RPMI-1640 medium with FBS at different concentrations up to 0.5 mg/mL was added and incubated for 24 h at 37 °C. Subsequently, the suspension was removed, followed by washing in PBS for 3 times. For evaluation of cell viability, PBS was replaced by FBS-free RPMI-1640 (200 μL), supplemented with 20 μL of Cell Counting Kit-8 (CCK-8) solution and incubated for 1.5 h at 37 °C. Absorbances of media and suspensions at 450 nm were subsequently measured using a microplate reader (Thermo, Varioskan Flash, Vantaa, Finland) and the cell viability was expressed as
(6)$$\mathrm{Cell}\;\mathrm{viability}\left(\%\right)=\frac{A_{with\;micelles}-A_{medium\;only}}{A_{CCK-8\;only}-A_{medium\;only}}\times 100\%$$
in which *A_with micelles_* and *A_medium only_* refer to the absorbances at 450 nm of CCK-8 supplemented medium with cells and micelles and CCK-8 supplemented medium without cells and micelles, respectively. *A_CCK-8 only_* refers to CCK-8 supplemented medium with cells but without micelles. Cell viability was measured in six-fold from one cell culture.

### 2.9. Subcutaneous Infection Model

Seven-week-old, female BALB/c mice (18–20 g each) were purchased from Vital River Laboratory Animal Technology Co. (Beijing, China) and housed at the Model Animal Research Centre of Soochow University (Suzhou, China). All experiments were approved by Jiangsu Provincial Association of Laboratory Animals (reference number 220184009). After being housed for 3 days, the hair of the back of the mice was removed, and mice were tagged in the ear and a subcutaneous infection site was created in the dorsum of each mouse by injecting 60 μL of *S. aureus* Xen36 (1 × 10^9^ CFU/mL) suspension in PBS. This day is denoted as day-2. 

Two days after initiating infection, i.e., at day 0, mice were randomly divided into three groups of twelve mice and anesthetized by intraperitoneal injection of an aqueous solution of 4 wt% chloral hydrate (8.25 mL/kg body weight). Next, 200 μL of PDA-ICG/PEG-PAE micelles or PDA-ICG/PEG micelles in PBS (1 mg/mL) were administered by tail-vein injection. As a control, mice were injected with 200 μL PBS. At day 1 after initiating infection, the initial bioluminescent intensity arising from the wound site was recorded using a bio-optical imaging system (Lumina III, Imaging System, Perkin Elmer, Waltham, MA, USA, exposure time of 30 s, medium binning 1 f/stop, open emission filter). Subsequently, each group of mice was split in two groups. One group received irradiation at the infection site with an 808 nm NIR laser (10 min, 1300 mW/cm^2^), while the other group was left without irradiation. 24 h after administration of the first injection, encapsulated photothermal nanoparticles were administered again and administration was repeated for three consecutive days. Bioluminescent images were recorded every other day, preceded by NIR irradiation (depending on the group considered). On day 8, all mice were sacrificed. For CFU enumeration, tissue samples from the infection sites were excised, weighted and put into a tube with 1 mL of PBS. After homogenization (Diunce Tissue Grinder), 2 mL of PBS was added to each sample and vortexed for 1 min. Next, samples were serially diluted and plated on TSB-agar with kanamycin for growth of *S. aureus* Xen36. After culturing at 37 °C for 24 h, numbers of CFUs were enumerated.

### 2.10. Biosafety Analyses 

In order to evaluate the short-term biosafety of the PDA-ICG/PEG-PAE micelles, 200 μL of PBS or a micelle suspension in PBS (1 mg/mL) was injected into the tail-vein of a healthy mouse. Injection was repeated for 3 days consecutive days. After 14 days, blood was collected through the eye, mice were sacrificed and internal organ tissues collected from the heart, liver, spleen, lungs and kidneys. Three mice were used in each group. Blood biochemistry was evaluated for liver and kidney function, including lymphocytes (Lymph), granulocytes (Gran), urea nitrogen (UREA), albumin (ALB), alanine transaminase (ALT), aspartate transfaminase (AST), levels of white blood cells (WBC), red blood cells (RBC), mean corpuscular volume (MCV) and mean corpuscular hemoglobin concentration (MCHC). 

For histological analysis, organs were fixed in neutral buffered formalin for 24 h and dehydrated in a graded series of ethanol (75%, 85%, 90%, 95%, 100%) using a dehydrating device (DIAPATH, Donatello, Martinengo, Italian). After dehydration, samples were embedded into paraffin. Slices of 4 μm were made and put on slides and dried at 60 °C. Next, slices were dewaxed, stained with hematoxylin and eosin, sealed with neutral gum and light microscopically examined (Leica DM4000M, Mannheim, Germany).

### 2.11. Statistical Analyses

Differences were studied for statistical significance using one-way ANOVA after Bonferroni multiple comparison correction (GraphPad Prism v. 8.1.1, San Diego, CA, USA), accepting significance at *p* < 0.05.

## 3. Results

### 3.1. Characterization of PDA- and PDA-ICG Nanoparticles before and after Micellar Encapsulation

TEM micrographs show that PDA-NPs have a diameter of around 50 nm. Despite drying for TEM preparation, most nanoparticles appeared as single particles with very little aggregates due to the hydrophilicity of PDA-NPs (Figure 1a). However, after ICG coating, nanoparticles appeared in aggregates, attesting to the hydrophobicity created by the ICG coating (see also Figure 1a). Coating with ICG could furthermore be inferred from the development of a broad UV-vis absorption band due to ICG (see Appendix A for UV-vis absorption spectrum of ICG) in the spectrum of PDA-ICG-NPs that persisted upon micellar encapsulation (Figure 1b). In absence of ICG modification, PDA-NPs had a mean diameter of 60 ± 1 nm as measured using dynamic light scattering (Figure 1c), corresponding with the diameter of single nanoparticles in TEM micrographs. Modification of PDA-NPs with ICG increased the mean hydrodynamic diameter to 90 ± 1 nm due to aggregation of hydrophobized PDA-ICG-NPs. Micellar encapsulation of PDA-ICG-NPs further increased the mean hydrodynamic diameter of the nanoparticles to 125 ± 10 nm and 135 ± 14 nm for PEG and PEG-PAE micelles, respectively, likely as a result of ongoing aggregation of the hydrophobic nanoparticles during encapsulation (see also Figure 1c). Zeta potentials of PDA-NPs and PDA-ICG-NPs were negative at both pH 7.4 and pH 5.0 (Figure 1d). Micellar encapsulation reduced the zeta potentials to almost zero at pH 7.4, but only yielded a highly positive zeta potential at pH 5.0 after encapsulation in PEG-PAE micelles (see also Figure 1d), which is required for targeting negatively-charged [26] bacteria. 

Heat generation in suspension increased upon coating PDA-NPs with ICG (Figure 1e), while micellar encapsulation of PDA-ICG-NPs had no influence upon heat generation. Equilibrium was reached within 600 s of NIR irradiation. Accordingly, the photothermal conversion efficiency of PDA-NPs was calculated using Equation (4) (Appendix A) and found to increase from 33% for PDA-NPs to 47% upon coating with ICG, with no influence of micellar encapsulation. Heat generation was not affected over at least five on/off cycles of NIR irradiation (Figure 1f).

### 3.2. Photothermal Killing of Planktonic S. aureus by PDA-ICG-NPs before and after Micellar Encapsulation

Photothermal killing of staphylococci in suspension by PDA-ICG-NPs and encapsulated PDA-ICG-NPs upon NIR irradiation were significantly higher (*p* < 0.05, two-way ANOVA) than killing by PDA-NPs at concentrations below 0.5 mg/mL. Photothermal killing by PDA-NPs was less than 2 log-units and significantly smaller than by PDA-ICG-NPs up to a concentration of 0.2 mg/mL (Figure 2). At pH 7.4, there were no differences between encapsulation in PEG-PAE or PEG micelles at the highest concentration of 0.5 mg/mL applied (Figure 2b). At pH 5.0, however, staphylococcal killing by PDA-ICG/PEG-PAE micelles was significantly higher (*p* < 0.05, one-way ANOVA) already at low concentrations (<0.2 mg/mL; Figure 2a) than by PDA-ICG-NPs encapsulated in PEG micelles.

### 3.3. pH-Induced Penetration and Accumulation of S. aureus in a Biofilm-Mode of Growth by Photothermal PDA-ICG-NPs after Micellar Encapsulation

In order to compare penetration and accumulation, encapsulated PDA-ICG-NPs were first loaded with Nile red. Previously Nile red-loading has been demonstrated not to affect the properties of PEG or PEG-PAE micelles [15]. PDA-ICG-NPs encapsulated in PEG micelles were unable to target, penetrate, and accumulate in 48 h grown *S. aureus* biofilms in vitro, neither at pH 5.0 nor at pH 7.4 (Figure 3a). At pH 7.4, also PDA-ICG-NPs encapsulated in PEG-PAE micelles were unable to target, penetrate and accumulate in *S. aureus* biofilms in vitro, but at pH 5.0 clear accumulation was observed after 60 min exposure of the biofilm to nanoparticles encapsulated in PEG-PAE micelles (Figure 3b). Quantitative analysis of the images yielded the conclusion that PDA-ICG-NPs encapsulated in PEG-PAE micelles accumulated two- to three-fold better (*p* < 0.05, one-way ANOVA) in *S. aureus* biofilms at pH 5.0 than at pH 7.4 and, regardless of pH, better than PDA-ICG-NPs encapsulated in PEG micelles (Figure 3c).

### 3.4. Cytotoxicity, Hemolysis and Biosafety of Photothermal PDA-ICG-NPs after Micellar Encapsulation

Before carrying out in vivo experiments in mice, it was first established that photothermal nanoparticles encapsulated in PEG-PAE or PEG micelles were not cytotoxic or caused hemolysis in vitro over the concentration range applied (see Appendix A). Neither PDA-ICG-NPs encapsulated in PEG-PAE or in PEG micelles exerted any negative effects on the viability of L929 fibroblasts (Appendix A), indicative of the absence of cytotoxicity. Also, hemolysis by encapsulated PDA-ICG-NPs was similarly low for both types of micellar encapsulation (Appendix A).

Subsequently, biosafety of PDA-ICG/PEG-PAE micelles was evaluated in vivo (Appendix A). Histological analyses of major organ tissues demonstrated no abnormalities in mice injected with PDA-ICD/PEG-PAE micelles as compared with mice injected with PBS (Appendix A). Also, blood biochemistry in mice injected with PDA-ICG-NPs encapsulated in PEG-PAE micelles was similar as in mice injected with PBS (Appendix A).

### 3.5. Eradication of a Sub-Cutaneous S. aureus Biofilm in Mice by Photothermal PDA-ICG-NPs after Micellar Encapsulation 

Benefits of encapsulating photothermal nanoparticles in PEG-PAE vs. PEG micelles were explored in a sub-cutaneous infection model in mice using a bioluminescent *S. aureus* strain (Figure 4). Bioluminescence images clearly indicate the infection site in mice receiving different treatments (Figure 4b). Bioluminescence intensity arising from the infection site decreased significantly faster after tail-vein injection of targetable PEG-PAE encapsulated photothermal PDA-ICG-NPs combined with NIR irradiation than after injection of PEG encapsulated PDA-ICG-NPs and NIR irradiation (Figure 4c). Nevertheless, injection of non-targetable PEG encapsulated micelles combined with NIR irradiation showed a faster decrease in bioluminescence than observed upon injection of PBS. All three groups of mice demonstrated a similarly low decrease in bioluminescence in absence of NIR irradiation (Figure 4d). A similar pattern of superiority of targetable PDA-ICG-NPs encapsulated in PEG-PAE micelles combined with NIR irradiation follows from a comparison of the number of CFUs found in tissue excised from the wound site (Figure 4e).

## 4. Discussion

Photothermal nanoparticles of different sorts have been considered for many years as a new strategy for bacterial infection-control. A literature search on photothermal nanoparticles and bacterial infection yielded more than 200 hits over the past five years. Yet, it was only recently described that NIR irradiation of a small bacterial infection site exposed to photothermal nanoparticles can yield severe collateral tissue damage [13], limiting the possibilities of their potential clinical use as a therapeutic. Targeting NIR irradiation to an infection site requires extremely precise dosing of photothermal nanoparticles to prevent collateral tissue damage [13] as the heat generated spreads through adjacent tissue. Magnetic targeting of photothermal nanoparticles to an infection site has also been suggested [27,28,29], but the current state of clinically applied magnetic targeting instrumentation impedes targeting to micron-sized infectious biofilms [30].

Recent works on photothermal nanoparticles for bacterial infection control therefore focus on photothermal nanoparticles that have been modified with smart, chemical moieties that allow accumulation near an infection site. Preferably, smart photothermal nanoparticles should penetrate deep and evenly over the depth of an infectious biofilm. Most of the targeting moieties involve specific pathogen-recognitions, such as type IV pili or type III secretion protein for targeting *Pseudomonas aeruginosa* [31,32], vancomycin [33,34,35], anti-protein A IgY [36], endolysins or other specific bacterial receptors for targeting *S. aureus*, including MRSA strains [37,38]. Ideal therapeutic approaches should be based on non-specific bacterial targeting, at least until pathogen identification has been done in order to save valuable time during life-threatening infections. Targeting moieties that are able to find their own way through the body to an infectious biofilm have been suggested based on general properties of an infectious biofilm, most notably its acidity as compared with healthy, surrounding tissue [39]. Photothermal nanoparticles modified with Pep-DA [40], ciprofloxacin with conjugated carboxyl betaine [41], zwitterionic ammonium compounds [42,43] or polyaniline-conjugated glycol chitosan [44,45] become positively-charged in an acidic environment yielding electrostatic double-layer attraction with negatively-charged bacterial cell surface [46]. Often, however, synthesis is laborious [39] and multiple modifications on a single nanoparticle, like achieving stealth and pH-responsiveness, are difficult to balance [47,48,49].

The approach chosen here to create smart, self-targeting photothermal nanoparticles is entirely different and relies on encapsulation of highly photothermal PDA-ICG-NPs in PEG-PAE micelles. PEG-PAE micelles have balanced stealth and pH-responsiveness and have been demonstrated to find their own way through the body of a mouse from a tail-vein injection site to an abdominal biofilm within half an hour after injection as mediated by the charge reversal of PAE in an acidic environment without observable harm to mice [15,50]. Transportation through the blood circulation and tissue occurred in a biosafe manner, although biosafety was only demonstrated for the short-term. Unfortunately, long-term biosafety is impossible to address, since adverse effects of any new therapeutic, regardless for the control of infection or any other disease, might turn up many years later. As a disadvantage, however, PEG-PAE micelles can only be used as nanocarriers for hydrophilic substances [24], which we have overcome by making PDA-NPs hydrophobic with the use of ICG.

## 5. Conclusions

Encapsulation of photothermal PDA-ICG-NPs in stealth and pH-responsive PEG-PAE micelles yields effective targeting and eradication of an infectious biofilm upon tail-vein injection of encapsulated photothermal nanoparticles in mice. PEG with its stealth properties and the small negative charge of PAE ensured transport through the blood circulation, while charge reversal in the acidic environment of an infectious biofilm caused electrostatic double-layer attraction with negatively-charged bacteria in a biofilm. This accumulation of photothermal nanoparticles in the target biofilm, combined with localized NIR irradiation, prevented collateral tissue damage. Herewith, the application of photothermal nanoparticles as a new strategy for infection control has been brought closer to the clinic.

## Figures and Tables

**Figure 1 nanomaterials-11-03180-f001:**
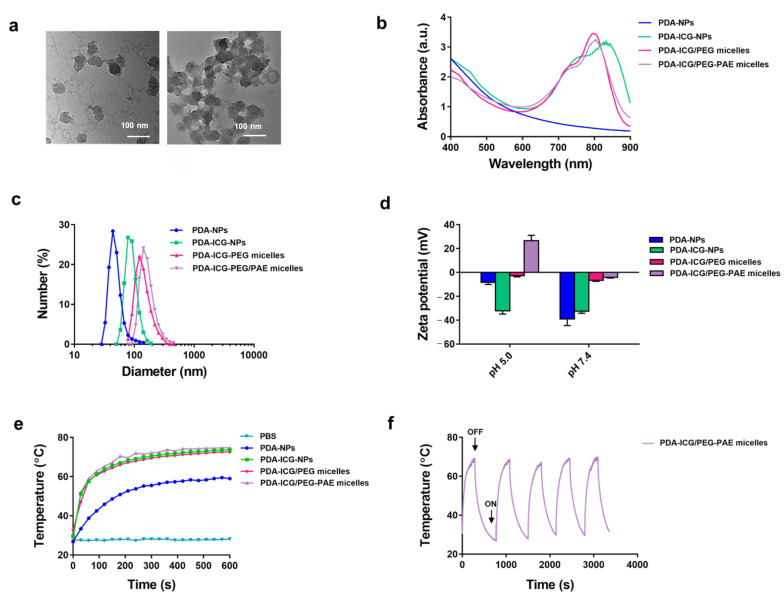
Characteristics, including photothermal properties, of photothermal nanoparticles before and after micellar encapsulation. (**a**) Transmission electron micrographs of PDA- and PDA-ICG-NPs. Note aggregation of hydrophobic PDA-ICG-NPs. (**b**) UV-vis absorption spectra of nanoparticles and PDA-ICG/PEG-PAE and PDA-ICG/PEG micelles (see Appendix A for a UV-vis absorption spectrum of ICG in water). (**c**) Diameter distributions of nanoparticles before and after encapsulation in PEG-PAE or PEG micelles, obtained using DLS in PBS. (**d**) Zeta potentials of nanoparticles in 10 mM phosphate buffer at different pH before and after encapsulation in PEG- and pH-responsive PEG-PAE-micelles. Data represent means ± standard deviation over triplicate experiments. (**e**) Temperature of nanoparticle suspensions with and without micellar encapsulation in PBS (250 µL, 0.2 mg/mL) as a function of NIR irradiation time (808 nm, 1300 mW/cm^2^). (**f**) Photothermal stability of PDA-ICG/PEG-PAE micelles as a function of time during repetitive NIR irradiation. See panel e for conditions applied. ON/OFF refers to the action of turning the NIR-laser on or off.

**Figure 2 nanomaterials-11-03180-f002:**
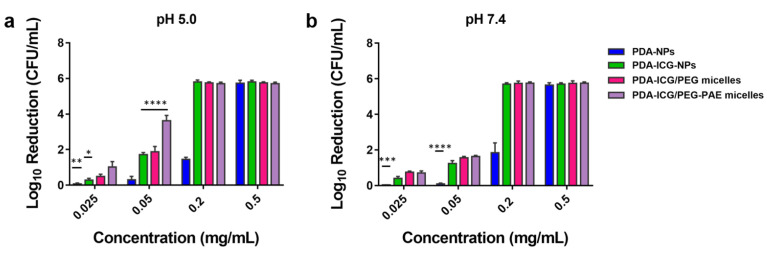
Photothermal killing of planktonic *S. aureus* Xen36 (3 × 10^6^ CFU/mL) in suspensions with different concentrations of photothermal PDA-NPs and PDA-ICG-NPs after micellar encapsulation. Heat generation occurred in a suspension volume of 250 µL upon 10 min NIR irradiation (808 nm) at 1300 mW/cm^2^. (**a**) Log-reduction of staphylococci in suspension due to photothermal killing at pH 5.0 compared to NIR irradiation in absence of nanoparticles or micelles. (**b**) Log-reduction of staphylococci in suspension due to photothermal killing at pH 7.4 compared to NIR irradiation in absence of nanoparticles or micelles. Data represent means with standard deviations over 3 experiments with separately prepared batches of nanoparticles and bacterial cultures. * indicate statistically significant differences between the data indicated by the spanning bars (two-way ANOVA; * *p* < 0.05, ** *p*< 0.01, *** *p* < 0.001, **** *p* < 0.0001).

**Figure 3 nanomaterials-11-03180-f003:**
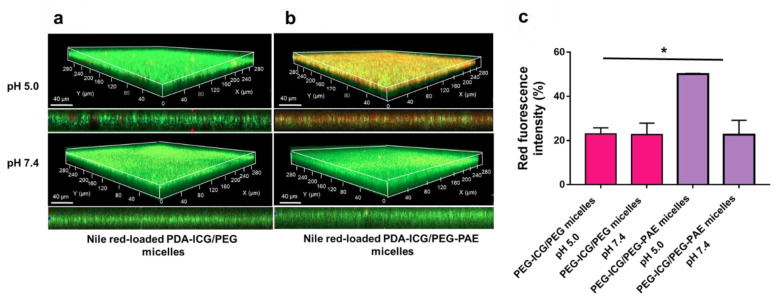
Targeting of *S. aureus* Xen36 biofilms by photothermal PDA-ICG-NPs after micellar encapsulation at pH 5.0 and 7.4. (**a**) CLSM micrographs of targeting, penetration and accumulation of encapsulated PDA-ICG-NPs into 48 h grown *S. aureus* biofilms, exposed for 60 min to PDA-ICG-NPs encapsulated in Nile red-loaded PEG micelles in suspension (0.5 mg/mL) for 60 min at different pH. (**b**) Same as panel a, now for encapsulation in PEG-PAE micelles. (**c**) Percentage red-fluorescence intensity upon accumulation of PDA-ICG-NPs encapsulated in Nile red-loaded micelles in *S. aureus* at pH 5.0 and pH 7.4. The percentage red-fluorescence intensity was expressed relative to the total (red and green) fluorescence intensity. Data represent means with standard deviations over 3 experiments with separately prepared batches of nanoparticles and bacterial cultures. * indicate statistically significant differences between the data indicated by the spanning bars (one-way ANOVA; * *p* < 0.05).

**Figure 4 nanomaterials-11-03180-f004:**
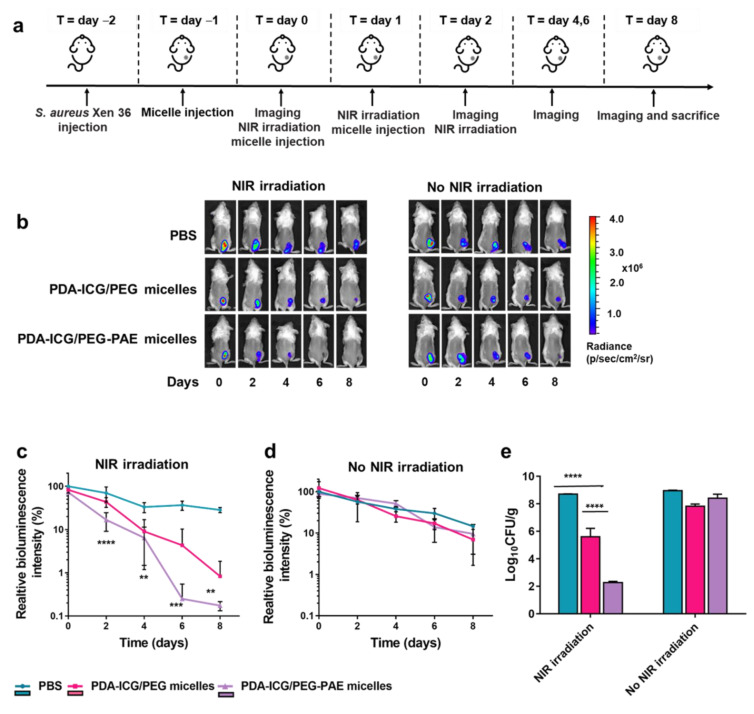
Eradication of a sub-cutaneous *S. aureus* Xen36 biofilm in mice after tail-vein injection of photothermal PDA-ICG-NPs encapsulated in micelles and in absence and presence of NIR irradiation. (**a**) Experimental scheme. (**b**) Time series of bioluminescence images of mice taken after injection of PBS (200 µL) or a suspension of micellar encapsulated photothermal nanoparticles in PBS (1 mg/mL) in absence or presence of NIR irradiation. (**c**) Bioluminescence intensity arising from the infection sites as a function of time after initiating injection and NIR irradiation. Bioluminescence intensity was expressed in percentages relating to the bioluminescence intensity at t = 0. (**d**) Same as panel c, now in absence of NIR irradiation. (**e**) Numbers of CFU/g tissue excised from around the infection site at sacrifice. Data represent means with standard deviations over 6 mice in each group. * indicate statistically significant differences between the data indicated by the spanning bars (one-way ANOVA; ** *p* < 0.01, *** *p* < 0.001, **** *p* < 0.0001).

## Data Availability

All data are available upon email request.

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
