# Peer review of "Encapsulation of Photothermal Nanoparticles in Stealth and pH-Responsive Micelles for Eradication of Infectious Biofilms In Vitro and In Vivo"

_nanomaterials, 2021, doi:10.3390/nano11123180_

Round 1

Reviewer 1 Report

In this study, the author proposed PEG-PAE-micelle encapsulated PDA-ICG-NPs for an improved anti-biofilm activity and limited cytotoxicity. Overall this manuscript provides solid experimental data for the proposed NPs delivery platform. However, the manuscript requires a big improvement on the language itself. Also, some of the studies is missing control groups. A couple of examples here:

  1. Starting line 48: …infection control has its own advantages and disadvantages. One of the major advantages is the broad-spectrum bacterial killing by photothermal nanoparticles on a non-antibiotic basis. However, comparing to photothermal treatment of centimeter-sized tumors, irradiation of a micrometer-sized infection site will usually cause collateral tissue damage due to the heat generated by photothermal nanoparticles.
  2. Line 54-55: please rephrase
  3. Line 56-57: it’s confusing with the single side bracket “)” after [6] and [9]. The line also needs to be rephrased. e.g., Different materials such as … have shown capability of heat generation upon NIR irradiation, among which PDA-NPs are easy to prepare and possesses excellent biocompatibility and biodegradability.
  4. Line 64: , however,
  5. Line 101: please be consistent with the past tense/present tense. e.g., The synthesis of PDA-NPs has been previously reported and was briefly repeated in …
  6. Line 137: it would be better to include the explanation here why the pH was chosen at 5.0.
  7. Multiple studies such as Figure 2, 3, S3, are missing control group (e.g., without any NP treatment, PBS control for CFU/mL, etc.)
  8. Line 415: it was not until recently that NIR irradiation …
  9. Line 425: Even more preferably…
  10. Line 452: delete “Concluding,”

Author Response

In this study, the author proposed PEG-PAE-micelle encapsulated PDA-ICG-NPs for an improved anti-biofilm activity and limited cytotoxicity. Overall this manuscript provides solid experimental data for the proposed NPs delivery platform. However, the manuscript requires a big improvement on the language itself. Also, some of the studies is missing control groups. A couple of examples here:

  1. Starting line 48: …infection control has its own advantages and disadvantages. One of the major advantages is the broad-spectrum bacterial killing by photothermal nanoparticles on a non-antibiotic basis. However, comparing to photothermal treatment of centimeter-sized tumors, irradiation of a micrometer-sized infection site will usually cause collateral tissue damage due to the heat generated by photothermal nanoparticles.

ANSWER: English language use has been corrected by a native English speaker, as suggested by this reviewer.

  1. Line 54-55: please rephrase

ANSWER: The sentence has been rephrased.

  1. Line 56-57: it’s confusing with the single side bracket “)” after [6] and [9]. The line also needs to be rephrased. e.g., Different materials such as … have shown capability of heat generation upon NIR irradiation, among which PDA-NPs are easy to prepare and possesses excellent biocompatibility and biodegradability.

ANSWER: The single bracket was a mistake and has been removed. The sentence has been rephrased, as suggested by this reviewer.

  1. Line 64: , however,

ANSWER: Comma has been added.

  1. Line 101: please be consistent with the past tense/present tense. e.g., The synthesis of PDA-NPs has been previously reported and was briefly repeated in …

ANSWER: Thank you for noticing, we have checked the entire manuscript on the use of past and present tenses.

  1. Line 137: it would be better to include the explanation here why the pH was chosen at 5.0.

ANSWER: We have chosen two pH values, pH 7.0 to mimic physiological pH conditions and pH 5.0 to mimic acidic pH conditions in a biofilm. We have added this information to section 2.4.

  1. Multiple studies such as Figure 2, 3, S3, are missing control group (e.g., without any NP treatment, PBS control for CFU/mL, etc.)

ANSWER: The reviewer may have missed the fact, that we did include controls. For the data in Figure 2, we have used NIR irradiation without nanoparticles or micelles as a control and reductions were reported relative to this control. We have rewritten the caption for improved clarity.

In Figure 3, we solely show penetration and accumulation of the two different types of micelles with PDA-ICG-NPs after Nile red loading at pH 5 and pH 7.4 in order to demonstrate the benefit of PEG-PAE encapsulation versus PEG encapsulation.  

In the caption of Figure S3a it was mentioned that ‘Viability of fibroblasts incubated in medium only was set at 100%, implying medium was used as a control here. Similarly, in Figure S3b ‘Hemoglobin absorption of cells exposed to water was set at 100%’, implying water was used as a control. We have added this information to the figure for improved clarity.

  1. Line 415: it was not until recently that NIR irradiation …
  2. Line 425: Even more preferably…
  3. Line 452: delete “Concluding,”

ANSWER: Thank you for these suggestions. We have implemented all suggestions.

Reviewer 2 Report

The review comments are attached for the editor & authors.

Author Response

photothermal process basically accompanying with EM radiation and the final thermionic radiation (heat) induced progression. What’s the selection of the optimized radiation energy for the soft tissue like-biomaterials? Author’s must have to brief.

ANSWER: In a previous paper (Gao, R. et al. Nanomedicine 2021, 32, 102324, doi: 10.1016/j.nano.2020.102324), we have determined the optimal conditions as radiation energy and period for killing bacteria. Similar settings have been used in this paper. We have added this information to section 2.4.

What is fundamental photothermal aspect of in vivo & in vitro method for biogenic materials? What’s the key factors for biomaterials interaction and their effect for radiation effect at room temperature?

ANSWER: We consider a discussion of fundamental aspects of photothermal effects beyond the goal of this manuscript and as distracting from our goal. Hence we prefer not to follow up on this suggestion.

Specify the biofilms formation for the polymeric & biomaterials combination? Explain. What type of physio-chemical bonding play a role in the present studies? Chemical kinetic studies are missing in the work., future authors need to address.

ANSWER: Similar as above, we consider discussion of bonding mechanisms of biofilms to biomaterials beyond the goal of this manuscript. Also, we do not understand what is meant by “the biofilms formation for the polymeric & biomaterials combination” and missing “chemical kinetic studies”.

Pathogens i.e., bacterial killing by photothermal nanoparticles is the standard process nowadays? Justify.

ANSWER: The PDA-NPs encapsulated nanoparticles are very promising because they are self-targeting to an infectious biofilm. The low penetration of NIR in tissue is limiting the clinical use of these encapsulated photothermal PDA-NPs, while the heat generated may be excessive unless nanoparticles are carefully dosed. We have added this consideration to the Discussion. 

PDA-NPs in mixed shell polymeric micelles, composed of stealth poly-ethylene glycol (PEG) is stable in ambient temperature? Any alternative materials suggestion needs to address in the revised version.

ANSWER: With all due respect, we do not understand this comment, that is composed of two sentences: (1) PDA-NPs in mixed shell polymeric micelles, composed of stealth poly-ethylene glycol (PEG) is stable in ambient temperature? This is a question mark put behind an observation reported by us, but we fail to regard this as a question. (2) Any alternative materials suggestion needs to address in the revised version. Here, we do not understand what is meant by “needs to address in the revised version”.

Biofilm Penetration and Accumulation of Photothermal Nanoparticles after Micellar Encapsulation mechanism need to be explore.

ANSWER: We believe that in section 3.3 we have done exactly this (see Fig. 3), while mechanisms of penetration and accumulation are discussed in the last paragraph of the discussion.

Which is the best mode of Biosafety Analyses existing in universally? How do measure the biotoxicity? Explain.

ANSWER: With all due respect, we do not feel competent to claim to know “the best mode of biosafety analyses”. Biosafety can be measured in various ways, but the bottom line is that the animals should cure from a disease without any side-effects, on the short and long term. We studied short term effects by injecting micelles in a healthy mouse and analyzed internal organ tissue and biochemical blood parameters, as common in the literature. Experimentally, long term biosafety is hard to measure (and therefore never done), and a comment on this has been made in the discussion of the revised manuscript. Also additional information on the method used has been given (section 2.10).

Author have followed the standard ethical procedure?

ANSWER: Yes we have followed the standard ethical procedure and this information has been described in section 2.9.

Photothermal therapy does it require for oxygen or any reactive atmosphere condition to interact with the target cells or tissues?. Clarify

ANSWER: Photothermal therapy is based on the absorption of light by the PDA-ICG-NPs which is converted to heat, which is released in the biofilm and can be used under all conditions. The PDA-ICG/PEG-PAE micelles are pH responsive and become positively charged at low pH as in a biofilm, which will give electrostatic interactions with the negatively charged bacteria. This has been described in the Introduction and Discussion sections.

Current studies also show that photothermal therapy is able to use longer wavelength light, which is less energetic and therefore less harmful to other cells and tissues. What’s the best radiation (incident) wavelength light to optimize these kinds of biomaterials?

ANSWER: We have no scientifically founded opinion to address the question what “the best radiation wavelength is” and prefer not to speculate on this.

Author’s need to follow the journal standard format such as text, font, grammatical/syntax error etc have to carry out. Asides, the paper need to cite the following published journal in the introduction and suitable part of the revised manuscript. The suggested paper as follows: Biotechnology Letters, 2020, 42(5), pp. 853–863; Nanoarchitectonics in Biomedicine, 2019, pp. 585–618; International Journal of Pharmaceutical Sciences Review and Research, 2016, 40(1), pp. 318–323, 58 ; International Journal of Engineering and Advanced Technology, 2019, 8(6), pp. 3684–3687 ; Materials Today: Proceedings, 2019, 36, pp. 559–565. Therefore, the manuscript needs to be revised to publish in the same journal with Minor Revision after the above scientific, technical, and necessary suitable (mentioned) citing references.

ANSWER: We like to thank the reviewer for the many suggested references. We would be happy to include them but in our interpretation, these papers deal with topics that are highly remote from photothermal therapy. Therefore we have been unable to logically include them in the revised manuscript.

To illustrate this:
Biotechnology Letters, 2020, 42(5), pp. 853–863. This paper is about imaging of E. coli with ZnO- and Au-NPs and not about photothermal therapy or targeting of bacteria in a biofilm.

Nanoarchitectonics in Biomedicine, 2019, pp. 585–618. This book chapter is about tissue regeneration using polymer-based calcium phosphate scaffolds and not about photothermal therapy or targeting of bacteria in a biofilm.

International Journal of Pharmaceutical Sciences Review and Research, 2016, 40(1), pp. 318–323, 58. This paper is about targeting of diseases and not about photothermal therapy or targeting of bacteria in a biofilm.

International Journal of Engineering and Advanced Technology (IJEAT) ISSN: 2249 – 8958, Volume-8, Issue-6, August 2019. This paper is about Cobalt nanoparticles as an antibacterial tool and not about photothermal therapy or targeting of bacteria in a biofilm.

Materials Today: Proceedings, 2019, 36, pp. 559–565. This paper is about the preparation of NiO nanoparticles and their antibacterial activity and not about photothermal therapy or targeting of bacteria in a biofilm.